# Immune Dynamics Involved in Acute and Convalescent COVID-19 Patients

Alexander Leonardo Silva-Junior [1,2,3], Lucas da Silva Oliveira [2,3], Nara Caroline Toledo Belezia [2,3], Andréa Monteiro Tarragô [2,4], Allyson Guimarães da Costa [2,4,5] and Adriana Malheiro [1,2,4,5,*]

1   Programa de Pós-Graduação em Biotecnologia, Universidade Federal do Amazonas (UFAM), Manaus 69067-005, AM, Brazil
2   Fundação Hospitalar de Hematologia e Hemoterapia do Amazonas (HEMOAM), Manaus 69050-001, AM, Brazil
3   Centro Universitário do Norte (UNINORTE), Manaus 69020-031, AM, Brazil
4   Programa de Pós-Graduação em Ciências Aplicadas à Hematologia, Universidade do Estado do Amazonas (UEA), Manaus 69065-001, AM, Brazil
5   Programa de Pós-Graduação em Imunologia, Universidade Federal do Amazonas (UFAM), Manaus 69067-005, AM, Brazil
*   Correspondence: malheiroadriana@yahoo.com.br; Tel.: +55-92-99114-9478

**Abstract:** COVID-19 is a viral disease that has caused millions of deaths around the world since 2020. Many strategies have been developed to manage patients in critical conditions; however, comprehension of the immune system is a key factor in viral clearance, tissue repairment, and adaptive immunity stimulus. Participation of immunity has been identified as a major factor, along with biomarkers, prediction of clinical outcomes, and antibody production after infection. Immune cells have been proposed not only as a hallmark of severity, but also as a predictor of clinical outcomes, while dynamics of inflammatory molecules can also induce worse consequences for acute patients. For convalescent patients, mild disease was related to higher antibody production, although the factors related to the specific antibodies based on a diversity of antigens were not clear. COVID-19 was explored over time; however, the study of immunological predictors of outcomes is still lacking discussion, especially in convalescent patients. Here, we propose a review using previously published studies to identify immunological markers of COVID-19 outcomes and their relation to antibody production to further contribute to the clinical and laboratorial management of patients.

**Keywords:** SARS-CoV-2; inflammation; antibodies; adaptive immunity; immune hallmarks





## 1. Background

Coronavirus disease 2019 (COVID-19) is an infectious disease caused by a betacoronavirus, reported as the pathogen of severe acute respiratory syndrome coronavirus 2 (SARS-CoV-2). It was first described in 2019, related to transmission from wild animals, in the province of Wuhan, China. Cases rapidly increased around the world due to its easy transmission via aerosol; however, other fluids, such as urine and saliva, as well as surfaces (paper, wood, and metal) for more than 4 days [1–4], were suggested as transmission routes, but conclusive evidence is still needed.

After exposure to the virus, the respiratory tract is the main tissue affected by angiotensin-converting enzyme (ACE2) expression, the human's receptor to the viral spike protein. After binding, SARS-CoV-2 enters the target cell, thereby establishing infection. The human first line of defense plays an important role in immune recognition and management of the disease. The viral protein receptor is expressed mainly in pulmonary, cardiac, renal, and lipid-rich tissues, and symptoms are represented by cough and fever; in severe cases, symptoms may evolve to pneumonia and death. Dynamics of extrinsic factors, such as age and comorbidities [5–7], together with intrinsic factors, such as immune response, have been

demonstrated as key factors in a patient's clinical outcome during the acute phase, in their post-COVID-19 symptoms, and in their protection from new variants [8–10].

The innate and adaptive immune responses contribute to viral clearance and to the production of specific antibodies. Many studies have evaluated and characterized the immunity imbalance in patients with active disease, showing a hyperinflammatory response in severe patients, led by a cytokine storm that causes more disadvantages than benefits in the resolution of the disease [11–16]. However, few papers have proposed evaluating those patients who had a favorable outcome, known as convalescent patients.

Many individuals may present long-term symptoms, named post-acute COVID-19 (starting 3–12 weeks from the acute phase) and post-COVID-19 syndrome (or long COVID, starting more than 12 weeks from the acute phase), which highlights the damage caused by the acute phase, as well as the risk of adverse effects and death [17,18]. Leukocytes are responsible for acute dynamics, as well as the production of markers of immunity. Although leukocytes play many different roles against SARS-CoV-2, they are also a key factor in tissue repair and convalescence [19,20]. Higher levels of total leukocytes are described, but the immune cell and soluble protein profile changes during infection (active COVID-19) between mild and severe patients, especially to predict the consequences in convalescence [21–23].

Here, we propose a concise review of immune aspects in acute COVID-19 patients, as well as contribute to the understanding of the immune dynamics during active disease and the contribution to the convalescent stage. Comprehending the process involved can promote better clinical guidelines, identify better hallmarks, and improve the patient's quality of life.

## 2. An Overview of Immunology in the Acute Phase of COVID-19

The establishment of a SARS-CoV-2 infection triggers the host's immune response to recognition, inflammation, and viral clearance. Previous studies have highlighted the prevalence of neutrophils and monocytes as the first immune cells to migrate to the infectious site through the stimulus of chemokines and the expression of adhesion molecules by endothelial cells, acting as a key factor in innate immunity [24,25]. The recognition of microorganisms and infected cells is mediated by pattern recognition receptors, especially Toll-like receptors (TLRs), NOD-like receptors (NLRs), and others related to intermediate intracellular mechanisms of activation that contribute to the effects observed during the immunological response.

### 2.1. NETosis Plays a Pivotal Role in COVID-19 Pneumonia Severity

Neutrophils act mainly by phagocytosis, which occurs via the inclusion and digestion of components into intracellular organelles, mediated mainly by enzymes. In addition, neutrophils produce inflammatory mediators, such as reactive oxygen species (ROS), which contribute not only to the activation of other immune cells, but also to cell recruitment to the local site of infection [21,26]. Although cytokines and other proteins have been described in terms of neutrophil interaction in the immune system, their relationship with COVID-19 severity remains scarcely known. Neutrophils play an important role in innate and adaptive responses since it is the first cell to reach the inflammatory site, where they can recognize the pathogen, digest it, and promote an immune response through the release of inflammatory cytokines [27].

The absolute neutrophil count (ANC) has been reported as an important severity mediator among COVID-19 patients, along with lymphocyte count. Patients who required intensive care unit (ICU) admission for pneumonia caused by COVID-19 or who developed the severe form of the disease had higher values of ANC and/or neutrophil-to-lymphocyte-ratio (NLR) [6,10,24,25,28–35]. NLR determination is an easy and cost-effective test to perform in clinical practice, and it has shown a significant improvement in the stratification of COVID-19 patients at hospital admission [21,30,36,37], as well as its potential as a

prognostic factor for COVID-19 outcome [34,37–40], especially in those with comorbidities, such as type 2 diabetes, hypertension, and ischemic heart disease [25,32,41].

Relative values expressed as a percentage of neutrophils, related to absolute leucocyte count, did not demonstrate a significant difference in acute patients, even among individuals with confirmed SARS-CoV-2 infection and those patients exposed [21,42], emerging as a non-recommended parameter in clinical and laboratory COVID-19 management.

The acute phase is marked by inflammatory and inhibitory cell surface markers, such as CD63, CD64, CD117, and CXCR3, driven mainly by CXCL8 and G-CSF [42–46]. The participation of immature neutrophils marked by CD10$^+$ and CD16$^{low}$ is prominent in the severe form, with acute respiratory distress syndrome [28,47] and ICU patients close to discharge, when compared to moderate and mild patients, driven by G-CSF [43]. Due to the urgency of inflammatory mediators, the cell phenotype profile shows a 'shift to the left', with the presence of immature neutrophils, although it is unknown whether they are immunosuppressive or pro-inflammatory. A positive correlation of immature neutrophils was seen with inflammatory markers of IL-6, IL-1ra, CXCL8, CXCL10, CCL3, CCL4, and vascular endothelial growth factor [12,43,48,49].

CD11b, another important neutrophil marker related to adhesion to alveolar macrophages [50], was demonstrated to be controversial under COVID-19 disease activity [12,27,51], but was associated with prolonged viral replication, being significantly reduced in those with a poor outcome [51]. From the acute to convalescent stage, this marker was shown not to suffer a significant difference [42].

The expression of activated markers and adhesion molecules is important for a better understanding of neutrophil physiology and further therapeutic strategies. Thus, immature neutrophils demonstrated greater participation in COVID-19 disease, potentially due to their regulatory function, the tissue healing process, and the low expression of adhesion markers, which contribute to their maintenance in peripheral blood [12,49,52].

Severe stages of the disease are characterized by hypoxemia, which was demonstrated to activate transcriptional factor HIF-1$\alpha$ (hypoxia-inducible factor 1$\alpha$), responsible for further production of an inflammatory profile, guided mainly by IL-1$\beta$, IL-6, and IL-8 (CXCL8) [29,53] (Figure 1). Other molecules, such as platelet-derived factor 4 (PF4) and CCL5 chemokine, can trigger neutrophil activation and promote mechanisms that may aggravate the condition of COVID-19 patients [54].

The participation of chemokines and cytokines is still not clear; however, their participation can recruit neutrophils to local activity. When infected by SARS-CoV-2, the lungs express chemokines CXCL1, CXCL2, CXCL3, CXCL5, CXCL8, and CCL20 that induce neutrophil chemotaxis from blood vessels to the lungs [53,55]. Furthermore, cytokines and chemokines in COVID-19 patients are guided mainly by the inflammatory process, involving CCL2, CXCL10, CXCL8, IL-6, and tumor necrosis factor (TNF) [53]. These molecules are important acute inflammatory mediators, which drive chemotaxis, increase adhesion molecules, and induce positive immune regulation from neutrophils and other cells [53].

Novel neutrophil mechanisms have been an important focus of research. Neutrophil extracellular traps (NETs), nets composed mainly of histones and proteins (lactoferrin, cathepsins, elastase, and myeloperoxidase), along with cytoskeleton components and other plasmatic proteins, are among the major mechanisms of neutrophils to maintain homeostasis [56,57]. NET release and ROS production are based on the neutrophil maturation level. Immature granulocytes are reportedly lower during viral clearance, when compared to normal individuals, which suggests that it might reflect the recruitment of mature and efficient neutrophils to leave the circulation and migrate to the tissue [21].

NETs also have the property of activating inflammasome complex NLRP3, which is responsible for type I interferon production and inflammatory cytokines [58]. Additionally, there is a stimulus of adaptive immune response, endothelial injury, thrombosis mediated by the immune system, and occlusion of small vessels, which may worsen pneumonia experienced by COVID-19 patients [43]. On the other hand, the capture of microorganisms

and immune cells allows the maintenance of inflammatory damage and downregulates it, thus reducing the consequences of exacerbated inflammation.

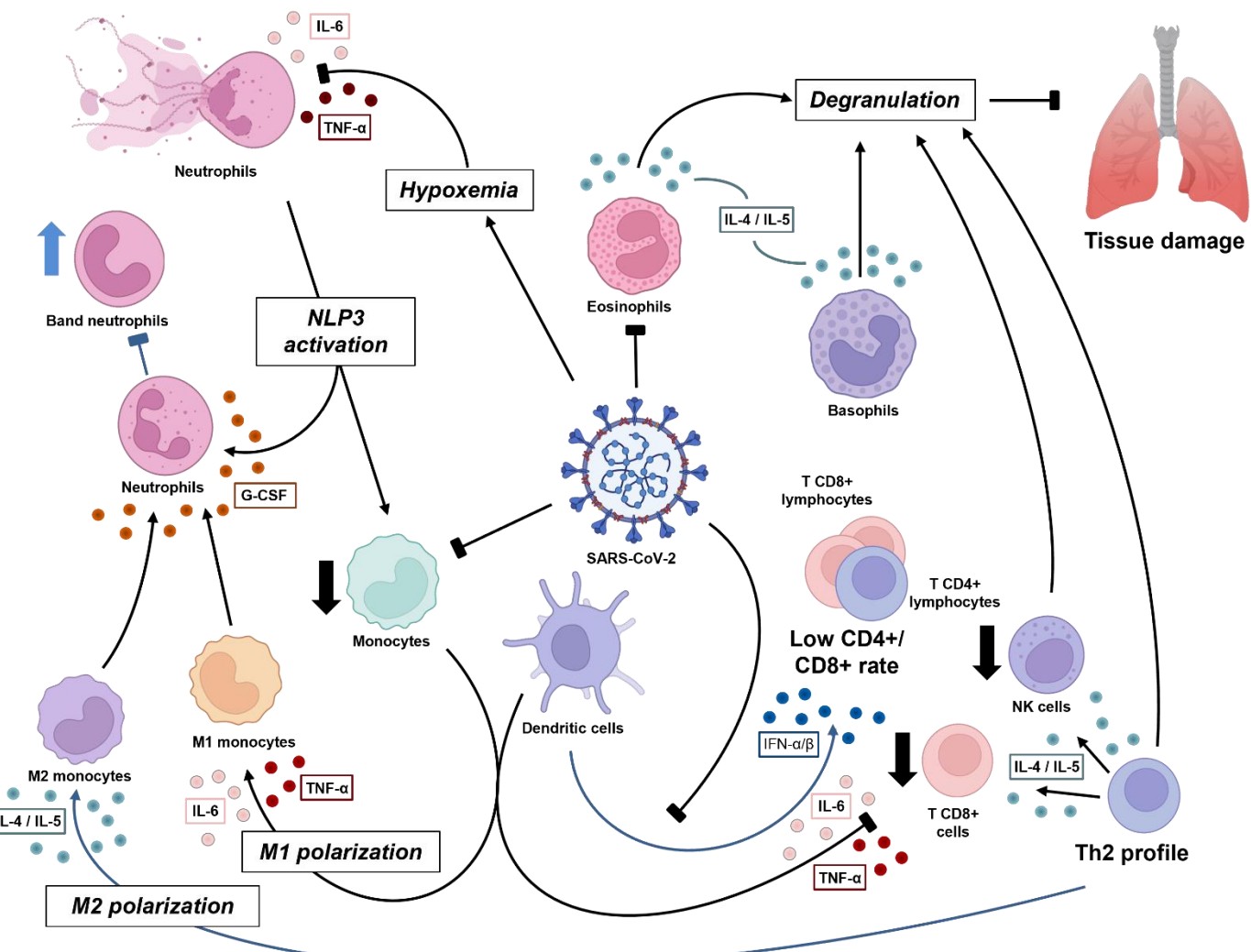

**Figure 1.** Dynamics of immune cell response against SARS-CoV-2 in acute phase. The virus induces pulmonary damage, which further causes hypoxemia. Both infection and tissue damage lead to inflammatory mediators, mainly interleukin-1β (IL-1β), CXCL8, and IL-6, which further cause NETosis and promote positive inflammatory feedback via inflammasome activation in both neutrophils and monocytes. Subsequently, critical patients experience low monocyte count, in terms of both T CD8+ and natural killer (NK) cells, with M1 polarization (driven by cytokine storm), all related to a worse clinical condition. Instead, when the antiviral stage is established, there is sustained type I interferon (IFN) by dendritic cells, contributing to the Th2 profile (IL-4 and IL-5), which induces the degranulation process by lymphocytes (CD8+ and NK), eosinophils, and basophils, and an M2 polarization, which results in a better prognosis. The release of granules has good and bad effects, whereby they can damage viral and human infected cells, but may also damage normal tissue, characterizing the respiratory syndrome. Macrophages produce granulocyte colony-stimulating factor (G-CSF), which acts in bone marrow to mature leukocyte progenitors. This may lead to a 'shift to the left' and improve viral response.

Several studies have described the formation of NETs in COVID-19 patients, with increased rates of markers in patients who evolved severe complications, such as thrombosis and death, or in intubated compared to non-intubated patients and convalescents [43,54,57], even though patients with a mild form of COVID-19 express a different type of immune

response. NET components myeloperoxidase (MPO)-DNA, neutrophil elastase (NE)-DNA, and citrullinated histones H3 exhibited no significant difference in mild and ICU patients, but they were increased when compared to healthy individuals [12,34,59]. In contrast, higher values of neutrophil elastase were found among ICU patients on entrance, with a median reduction after 7 days and further discharge [43]. Ng et al. [60] suggested a positive correlation of NET production, poor outcome, and inflammatory markers, such as WBC, ANC, and inflammatory cytokines, in addition to increased rates of thrombosis in the COVID-19 group.

A higher concentration of NETs in the respiratory tract than in circulating plasma was also observed, and a direct correlation index to clinical illness severity score was established [54]. It is important to note that NET release induces direct and indirect effects in a systemic manner. Activation of the complement system due to the presence of tissue factor, along with the formation of aggregates in the pulmonary compartment, worsens the clinical condition of pneumonia [56,61].

The thrombosis effects caused by NETs can cause injury not only to the pulmonary tract, but also to the endothelial, renal, liver, and cardiac systems, and they may be involved in further organ failure. Individuals with previous chronic disease, characterized as groups at risk of COVID-19, may experience an increase in endothelial injury due to the intense activity of immune cells, cytokine storm, and inflammatory mediator production [44,54,56]. The participation of thrombotic factors, such as D-dimer and von Willebrand factor, has been reported as being more intense in severe COVID-19 patients; although few studies have described the direct correlation of NETosis and thrombotic molecules, a relationship among NET, platelet aggregates, and the state of thrombosis has already been proposed [44,54,60].

## 2.2. Improvement of Severe Cases Is Marked by Eosinophilia

Eosinophils are polymorphonuclear cells that play a role in innate immunity; they are characterized by granules extremely rich in inflammatory mediators, such as cationic proteins, peroxidase, hydrolase, and lysophospholipase. This lineage has been extensively studied in terms of parasite response and allergic diseases [62]. This cell profile also presents PRRs and produces cytokines, nitric oxide, and proteins, which contribute to viral clearance [63].

There is an important relationship between cytokines and eosinophil production. It is already known that IL-4 promotes the expression of adhesion molecules that further contribute to eosinophil adhesion to the endothelium, while IL-5 induces degranulation. Both cytokines are produced mainly by mast cells, basophils, and Th2 lymphocytes, which are extremely important in allergic responses. The degranulation process is commonly used in parasites due to membrane damage, but it can also damage the host tissue [62].

Eosinopenia ($<40/mm^3$) was reported in patients admitted to the ICU, where it was suggested as a prognostic factor for poor outcomes [7,8,21,63–67]. In severe patients, $CD8^+$ T cells contribute to eosinophil proliferation via the production of IL-5; however, due to exhaustion on the first cell, the IL-5 level may suffer interference, which might be a cause of eosinopenia at the beginning of the disease course [36]. Around 20 days after hospital admission, the absolute eosinophil count (AEC) usually exceeds $1500/mm^3$ in patients ready for discharge. This eosinophilia commonly lasts around 5 days and is correlated with a reduced mortality rate [8,53,64,68,69].

Asian race/ethnicity with eosinophilia was a predictor for a shorter hospital stay, while other races/ethnicities showed no significant difference [63]. Other factors that have been proposed to contribute to no resolution of AEC are higher age, alcohol abuse, tobacco use, hypertension, diabetes mellitus, chronic pulmonary disease, chronic kidney failure, comorbidities, previous use of corticoids, and initial symptoms of normal cough, dyspnea, arthromyalgia, asthenia, and saturation >95% [8]. Some studies with asthmatic patients presented the same pattern, in addition to reporting the protective aspect of eosinophilia and a Th2 cytokine profile on disclosure of COVID-19 patients with asthma [63,70].

This cell lineage was also seemingly connected to severe symptoms, with a few patients with normal AEC experiencing fever, fatigue, shortness of breath, and inflammatory infiltrates at hospital admission, in addition to increased rates of aggravation [69], related to the natural killer (NK) T-cell response and further eosinophilic lung inflammation [26].

Sustained eosinopenia was found in severe cases and in patients with cytokine storm syndrome [28,36], and three hypotheses have been suggested [69]: (1) production of corticosteroids by the adrenal during an acute response, which blocks the release of eosinophils from bone marrow and induces migration of eosinophils to the tissue, culminating in reduced eosinophils in circulation; (2) COVID-19 may cause damage to the bone marrow, which would also impact eosinophil count (hypothesis not fully elucidated); (3) upregulation of Th1 and Th2 cytokines by viral clearance promotes leukocyte migration to pulmonary tissue, resulting in a lower availability in peripheral blood. Although not widespread, eosinopenia might also be related to the infection of eosinophils by SARS-CoV-2, as previously demonstrated [9]. These issues can be highlighted, especially as the eosinophil count increases at the same rate as clinical improvement and viral load reduction.

Both eosinophils and neutrophils were found in the bronchoalveolar fluid extracted from patients with severe COVID-19, in addition to eosinophil cationic proteins, which confirms the importance of eosinophils in the local immune response against SARS-CoV-2. The number of eosinophils in the pulmonary tract can cause similar inflammation to acute eosinophilic pneumonia, with a previous observation of >25% of eosinophils in the lungs [26]. Although it is known that COVID-19 is the agent responsible for leukocyte recruitment to the pulmonary tissue, the role of eosinophils in viral clearance is not fully understood.

The surface markers on COVID-19 patients demonstrate an activate profile characterized by a lower expression of CD15, CD66b, and CD193 and a higher expression of CD62L, CD69, and CD147, compared to noninfected individuals. A comparison between moderate and severe patients revealed CD69$^+$ eosinophils in the latter, which might be related to decreased outcome, whereas CD66b, CD11b, CD11a, and CD24 are present in eosinophil membranes in moderate patients, influencing clinical management [28].

Activated eosinophils (CD69$^+$) exhibit a positive correlation with soluble inflammatory molecules in severe patients, such as IFN-$\gamma$, CCL2, CCL7, and CCL8. They play a key role in lung tissue infiltration, degranulation of neutrophils, clotting factor activation, molecule recognition, and extracellular matrix metabolization [28].

### 2.3. Granulocytes and Monocytes Management in Viral Clearance

COVID-19 pathology is guided by a Th2 cytokine profile, which is directly connected to the participation of eosinophils, basophils, and the local inflammatory response. The absolute basophil count (ABC) was found at lower levels during the initial course of the disease, while showing recovery over the course of illness [22]. An absence of ABC recovery was present in severe patients who required mechanical ventilation and who evolved to a fatal outcome, possibly being a biomarker for a poor outcome [51]. A negative relationship between basophil count and both severe and hospitalized COVID-19 patients demonstrates the importance of basophils in local viral clearance [71].

The evaluation of basophils during disease activity demonstrated an increased rate of CD131$^+$ (IL-3 membrane receptor) cells, CD11b, CD63, and CXCR4 [28] but a low expression of GM-CSF and IL-5 receptors [51]. Furthermore, the involvement of thrombotic events in severe cases, as well as immunomodulatory effects during the acute and chronic responses, highlights the important role of basophils in immunity against SARS-CoV-2 [71]. Basophils are involved in the hypersensitivity response, production of mucus, vaso-constriction, inflammation, and tissue damage, but more studies must be conducted to evaluate their effect in acute COVID-19 [62].

Monocytes, on the other hand, represent a subpopulation of leukocytes from the same precursor as neutrophils. They are known as agranulocytes, with a main function related to the recognition of pathogens and cell products by PRRs and subsequent phagocytosis.

The stimulus activates intercellular mechanisms that contribute to cytokine storm and migration to tissue.

These cells participate in both innate and adaptive immunity, acting in COVID-19 viral clearance or as antigen-presenting cells to combat the virus or induce antibody production, respectively. Monocytes are classified into three specific subtypes according to their cell surface proteins: classical ($CD14^{++}CD16^-$), inflammatory ($CD14^{++}CD16^+$), and patrolling ($CD14^+CD16^{++}$). It is important to highlight that other markers can also be used, such as chemokine receptors and cytokine production [72].

Monocytes also contribute to an interesting aspect of COVID-19 physiopathology, as they were previously shown to express ACE2, thus being influenced by the virus [73]. It was shown that prolonged viral infection (more than 10 days of positive RT-PCR tests from admission) can reduce ACE2 mRNA, as well as levels of soluble ACE2, instead quickly returning to normal in patients with a negative RT-PCR in less than 10 days [74].

The absolute monocyte count (AMC) is a low-cost measurement, and monocytosis/monocytopenia was previously related to hospital discharge. Those with higher AMC spent fewer days in hospital (15 days), while those with lower levels of AMC remained for a prolonged period (40 days) [73]. In addition to its functionality, low rates of monocytes were found in severe and in mechanical ventilation patients [21,40,51,75], further associated with age [33] and the presence of atypical and vacuolated monocytes [73]. It was suggested that these morphological changes come from the process of monocyte infection, as also observed in visceral leishmaniasis, but not other viral diseases [73].

Severe disease is marked by a lower rate of AMC when compared to mild cases [40,75,76]. Monocyte soluble markers show that, during acute COVID-19, there is a higher secretion of sCD14 and sCD163 when compared to normal individuals. sCD163 is correlated with the time elapsed from hospital admission, whereas sCD14 is correlated with several laboratory parameters, including IL-6 and C reactive protein (CRP). Accordingly, a few differences were observed between ICU and non-ICU patients, but patrolling monocytes produce less sCD163 and more sCD14 [23,49,77–79]. Corticoids were suggested as a factor interfering with monocyte activation and inflammatory pathways, although the CD163 receptor was increased in all subtypes of monocytes in severe conditions [10,75,80–82].

The participation of monocytes/macrophages in pulmonary inflammatory diseases has been reported, especially classical monocytes in asthma [83,84]. Some studies have reported higher levels of monocytes in peripheral blood with a further reduction in convalescent stage, whereas others have reported the opposite [21,85,86], with similar observations in bronchoalveolar fluid [26]. A delay in interferon signaling leads to the infiltration of monocytes into the pulmonary tract, thus inducing the production of inflammatory mediators that drive the response to cytokine storm, which results in positive feedback to leukocytes, contributing to tissue damage and regulation of antiviral cytokines. Coronaviruses mediate antiviral cytokines via translational mechanisms that are usually involved in viral clearance, such as type I interferon, which acts as an escape mechanism [87].

During cytokine storm syndrome, monocytes tend to reduce their quantitative circulating value, but increase the granularity and permeability of endothelial cells [36,85]. This might result in the expression of adhesion molecules and migration to other tissues. Although it is well established that cytokine storm is the main event that influences a worse prognosis, monocytes (and even dendritic cells) were suggested to not be the main producers of proinflammatory cytokines [88].

The infection stage involves the participation of $CD16^+$ monocytes, in contrast to a healthy status [73,79]. However, among COVID-19 subgroups, mild and severe stages presented increased rates of inflammation ($CD16^+$) and presenting ability ($HLA-DR^+$) when compared to critical patients [49,51,75,76,81]. Winheim et al. [89] demonstrated that, within the subpopulation of activated monocytes, there was participation mainly by classical monocytes. This inflammatory profile was demonstrated to be guided by higher levels of TNF-α, IL-6, IL-10, IL-2, IL-4, IL-13, IL-18, CCL3, CCL4, and CCL2, although only IL-6 had a significant positive relationship with $CD16^+$ monocytes and a negative relationship with

HLA-DR$^+$ monocytes [19,76,90–93]. This suggests that proinflammatory monocytes drive IL-6 production, which may be related to the proinflammatory state and low HLA-DR production in critical patients (Figure 1).

We must highlight that both extra- and intracellular mechanisms play an important role in SARS-CoV-2 clearance, especially inflammasome activation, as mentioned before, in neutrophils and monocytes. Interactions among viral RNA [94], NETs [43,58], and the dysregulation of calcium concentration [95] influence NLRP3 activation, mediated by the viral envelope protein. Upon stimulus, this induces the maturation of proinflammatory cytokines IL-1β and IL-18, which further stimulates IL-6 and TNF, promoting inflammation in the lungs. These events worsen disease progression, representing the major mediators of cytokine storm, and cause systemic dysfunctions, such as macrophage recruitment and leukocyte degranulation [96]. Moreover, severe patients experience an increased rate of activation of the NLRP3 and TXNIP inflammasome pathways [79].

The expression of surface markers demonstrates a predominance of M1 macrophages in critical patients, when compared to noncritical patients, due to the increased expression of CD80, higher production of IL-6, TNF-α, and TGF-β cytokines (although *M*1 macrophages are not the main source of these cytokines), and lower expression of MHC-II [73,82,88,97]. It is important to note that *M*1 macrophages show increased odds of tissue migration due to their rheologic properties, but also fewer acid granules, which contribute to viral RNA perseverance in the cell. *M*2 macrophages, otherwise, present more acid granules, which lead to viral RNA instability and further degradation [90,98]. Even though alveolar *M*2 macrophages (CD206$^+$) are also present during the immune response, when compared to the healthy status, the cytokine profile barely varied, with an IL-4/IL-13 balance to induce *M*2 macrophages [73,99]; however, a decrease in CD86$^+$ expression and MHC-II [88] was observed.

Dendritic cells (DCs) play an important role in viral clearance, antigen presentation from innate to adaptive immunity, and the capture of apoptotic/necrotic cells [100]. Immature DCs show an increased ability to recognize antigens, while mature DCs are important producers of IL-12, IL-1β, type I and type II IFN, IL-4, IL-10, and TNF-α. [101]. Their function in COVID-19 disease remains unclear, and their participation is mainly guided by inflammatory status. Some studies demonstrated a lower expression of c-KIT$^+$ in cDC1 and of plasmacytoid DCs in mild/moderate patients, together with a significant reduction in DC count. Moderate patients were marked by a higher expression of CD38. However, with the progression of disease severity, a lower participation of the inflammatory DC3 subset (CD163$^-$CD14$^-$) was reported, with an increase in the c-KIT receptor [49,81,102,103].

Patients both with and without neurological symptoms 4 weeks after disease onset demonstrate increased rates of DC density and mature DCs, when compared to a healthy status. However, differences were not observed between infected groups [104]. These data suggest long-term participation of DCs, whether the patient is symptomatic or not. Cell activation and inflammatory status are compromised by increased age, and senescence is a common factor of immunity, which, when related to COVID-19, seems to play a key role in response, potentially reflecting why newer patients experience disease differently [101].

During viral infection, plasmacytoid DCs (pDC) are important producers of type I IFN (especially IFN-α) via recognition of RNA by TLR7/8, once there is participation of the PD-L1$^+$CD80$^-$ DC population; however, their levels are diminished with severity [49]. Even though a phenotypic profile is prominent in asymptomatic patients, hospitalized patients display a phenotype characterized by PD-L1$^+$CD80$^+$. It is important to highlight the higher expression of CD86 in asymptomatic patients [49], demonstrating their greater ability to stimulate DCs, whereas hospitalized patients had a higher expression of CD80 [12,103] and lower HLA-DR [105].

During viral infections (including viruses other than coronavirus), viral escape can occur through interference with type I IFN production via antagonism of transcriptional factors. This was established by some studies describing a higher concentration of IFN-α and IFN-genes at the beginning of disease, with a subsequent reduction, even in patients

that experience a severe form, in comparison to mild patients [12,106,107]. IFN-α plays an important role in dendritic cell functionality during acute disease, even in convalescence. A normal level is not restored for more than 6 months after infection, and a positive correlation index between IFN-α cytokine and pDCs has been demonstrated, which might be related to the decrease in P1-pDCs in hospitalized patients [106]. Costimulatory molecules CD80 (B7-1) and CD86 (B7-2), expressed mainly by DCs, are capable of activating T cells and participating in the SARS-CoV-2 response. A lower circulating level of mDC and pDC CD86+ has been reported in the acute phase; however, their efficiency in activating T lymphocytes must be explored, as their dynamics in acute disease interfere with antibody titers during convalescence [106,108].

*2.4. Lymphocytes: When Adaptive Immunity Takes Place*

Lymphocytes are important cells related to antiviral regulation, maintenance of homeostasis, and inflammatory responses. They are commonly divided into T and B lymphocytes according to their functionality and cell surface marker expression; however, both classes participate in innate and adaptive immunity involving soluble proteins (mainly cytokines, chemokines, and antibodies) through cell-to-cell interactions [109].

The absolute lymphocyte count (ALC) parameter, whether alone or combined with neutrophil count, had a greater ability to estimate a worsening prognosis when low [110], especially in ICU patients [111,112] and in those who evolved to death [40]. The mechanisms that drive lymphopenia are still not clear, but some concerns have been raised, such as (1) lymphocyte infection by the virus, (2) cell migration from blood to tissue, and (3) damage to lymphoid organs (and further production of lymphocytes) [113,114].

Due to the intense participation of cytotoxic T (CD8+) cells, the CD4+/CD8+ ratio is lower in diagnosed patients and those under ventilation therapy [49,51], when compared to healthy individuals, guided by inflammatory TNF-α and IFN-γ cytokines as major contributors [41,111]. This suggests an inflammation process related to antiviral activity, especially due to the expression of suppressor marker CD57 and adhesion molecules (CD38 and HLA-DR) on cytotoxic lymphocytes [6,49]. It would be reasonable to think that a side-effect can be seen in terms of tissue damage, as inflammation induces degranulation activity from T CD8+ lymphocytes and can be harmful to regular tissues [115–118].

T lymphocytes can be divided into different subtypes, according to cytokine production and functionality. The cytokine environment is characterized by the main populations of Th1 (mediated by INF-γ), Th2 (IL-4 and IL-13), Th17 (IL-17A and IL-17F), Treg (IL-10, IL-35, and TGF-β), and Th follicular (IL-21). Other T lymphocyte subpopulations also participate, such as Th9 (IL-9) and Th22 (IL-22), but their role during COVID-19 disease is not well described [119]. Cytokine storm, known as the key factor underlying disease severity in COVID-19, is guided by IL-1β, IL-2, IL-6, IL-7, IL-8, IL-10, G-CSF, GM-CSF, CXCL10, CCL2, CCL3, IFN-γ, and TNF-α [118,120–122]. However, few molecules have been related to disease outcome, such as the perseverance of higher IL-1ra and chemokine (CXCL10, HGF, CCL3, CCL7, and MIG) concentrations in plasma associated with a worse outcome, whereas CXCL10 and CCL7 were shown to predict COVID-19 improvement [121].

Previous studies described an increase in T helper cells, albeit slowly, in mild disease, as well as in Treg cells (CD4+CD25high), B lymphocytes, and NKT cells [6,41,51,116]. The severity score presented a negative relationship with both CD8+ and CD4+ lymphocytes, as well as NK cells [5,118,123–125], which may be related to their degranulation ability. All subtypes (naïve, memory, and effector T CD8+ and CD4+ cells) had increased rates of both apoptosis and migration characteristics, suggesting an inflammatory functionality and local repairment [126]. In mild patients, the immune response is based on the nonconventional Th1 cell lineage; however, in patients who evolve to a severe condition, the Th2 profile also participates [26]. Regardless of symptoms and severity, lymphocytes express CD38, CD39, CD69, CTLA-4, HLA-DR, Ki-67, and PD-1 markers, potentialized mainly by T helper and cytotoxic memory cells [6,85,127]. This profile induces acute and local damage over the course of the disease, and some parameters may decrease with disease severity [128].

In moderate-to-severe patients, not only were T effector cells found to express granulysin, but NKT (CD160[+]) cells also produce more granzymes A, B, and H to solve viral infection, which persists for a longer time in critical patients [82,129]. A severe condition was marked by a significantly lower count of NK (CD56[+] and CD16[+]) cells [5], compromising cytotoxicity [115,130]. T CD8[+] memory cells and effector cells (CD45RO[+] and CCR7[−]) are influenced by the IL-15/IL-15RA axis [116,131,132], PD-1, and inhibitory receptors (NKG2A and NKG2D) during severity, which play a key role in the functional exhaustion of NK cells, senescence, and apoptosis [115,118,120,126,133]. Imbalance of the IL-15 axis may induce a significant reduction in NKT γδ cells (CD160[+]), which was previously proposed to promote rapid control of the disease via direct cytotoxicity, as well as induce cytotoxicity mediated by antibodies, similar to regular NK cells [79,126,129,131]. The increased rates of cytotoxic T cells and type I cytokines were described as biomarkers of a poor outcome in convalescence, which may be guided by the higher release of enzymes, inducing cellular damage in tissues [127,134].

During SARS-CoV-2 infection, a significant proportion of T γδ cells can be identified, when compared to healthy individuals, but no differences were observed between patients under ventilation and SARS groups [48,52,107,126,135,136]. Instead, these cells presented a higher expression of CD4[+] and CD25[+] markers [137], although mild patients also had increased levels of TCR γδ naïve and central memory cells compared to severe patients [135]. Mucosal-associated invariant T cells (MAIT) did not differ among COVID-19[+] patients, but correlated positively with patient's age and negatively with severity, whereby those with pneumonia, hypoxia, and ICU had lower levels [49,107,126,136], which were increased among patients who were discharged in <15 days, along with an increase in iNKT cells [107]. However, MAIT cells with the expression of the CD8 marker were increased in patients in the ICU, together with IFN-γ and granzyme B production. COVID-19 patients show higher CD69[+] expression on MAIT cells, which is further correlated with CRP, IL-18, and IFN-α levels, suggesting an influence on inflammatory markers [52,79,107]. Once these cells are in the minority in blood, their participation in the inflammatory status still requires further elucidation. Their reduction in severity might be related to the intense production of other innate immune cells with a more active functionality in inflammation than nonconventional T cells, although cytotoxicity is stimulated, even in minority cells.

Many studies have raised concerns about laboratorial and clinical markers, especially related to prognosis. Few bioinformatic studies have addressed this issue using acute patients; however, it remains unclear whether other hallmarks can be proposed, characterizing the various clinical profiles that patients may experience (asymptomatic, mild, severe, ICU, and convalescence). Considering the completeness of data, science faces a need to comprehend the mechanisms involved in immune dynamics underlying significant clinical changes. These questions must be addressed in future studies on coronavirus so as to construct a bridge between basic and clinical studies.

## 3. What Do We Know about Convalescence so Far

Convalescence is known as the stage after COVID-19 clinical recovery. This group is formed by patients who were diagnosed with COVID-19 (whether symptomatic or not) but did not evolve to death. Comprehending the immune system's behavior during the acute phase plays an important role in recognizing the key points related to clinical improvement, as well as in identifying novel hallmarks related to a better or poorer outcome, and identifying the dynamics in adaptive immunity [60,75].

In convalescents, a significant increase in activated neutrophil count (CD45/CD11b[+]) 28 days after clinical recovery was reported [21,27], although a reduction in CD64[+] neutrophils [45] and NETs was observed a few weeks [12,54] to months [60] after a positive SARS-CoV-2 RT-PCR test.

Activation markers on neutrophils have been assessed, and a lower metabolic function with few genetic materials has been observed in COVID-19 acute patients, compared to healthy individuals. However, this is even lower in convalescence, together with

higher rates of immature granulocytes, which might be related to a lower number of well-functioning neutrophils, as well as urgent granulopoiesis [21]. It was suggested that, after disease, there is an increased circulation of nonreactive neutrophils; however, after a period of time, the reactivity function re-emerges [21,82]. The neutrophil population seems to be controlled quantitatively, according to Rodriguez et al. [39], although evolution of the inflammatory profile from mild and severe patients to convalescence was not determined. This profile may be a consequence of the cytokine storm from SARS-CoV-2 infection to the convalescent period, especially related to the reduction in IFN-$\alpha$ levels over time since symptom onset [79].

Immature granulocytes are significantly increased during acute COVID-19, reaching even higher levels in convalescence [21], albeit not from the neutrophil lineage [49]. This might be due to the intense immune response to help fight against the virus, whereas, in convalescence, there is an urgent need for the immune replacement of functional cells.

Eosinophils tend to be reduced in bronchoalveolar fluid during the convalescent stage, as indicated by immature cell markers CD45$^+$/CD24$^+$/CD16$^{low}$ [26], whereas they tend to be increased in blood vessels [9,21,22]. A direct relationship between AEC and ALC was described, which could be a reasonable field to comprehend immune factors associated with a transient stage of acute complications and adaptive response [138].

Although not yet fully understood, convalescent patients ($\pm$14 days after clinical recovery) with a higher titer of antibodies also express a higher mean AEC and level of immature granulocytes [138]. Even though Vitte et al. [66] evaluated convalescent patients 28 days after symptom onset, where the median eosinophil count increased to a greater level than in patients in the mild group, the level only decreased in 3/19 severe patients, with the remainder presenting a better outcome. Eosinophils demonstrate an important correlation with inflammatory markers during all disease pathologies, considering specific markers of innate immunity following symptom onset, when considering the evolution of dendritic cells, T lymphocytes and monocytes, until the recovery stage [39].

Several studies have proposed a concise evaluation of eosinophils in convalescence; however, despite being classically known as elements of innate immunity, they were also demonstrated to participate in adaptive immunity. There is an urgent need to comprehend the biological mechanisms underlying this cell's behavior to improve therapeutic strategies.

The production of IL-6 and IL-4 by basophils in convalescence are associated with higher production rates of antibody against SARS-CoV-2. However, patients admitted to the ICU had a slight reduction in IL-6 levels, when compared to those who were not [39]. Although the mechanisms underlying the relationship among basophils, viral clearance, and antibody production are not fully understood, this cell lineage plays a pivotal role in viral load and local repairment.

During recovery (1–10 days after admission), the AMC tends to return to normality, and the level of activated monocytes is reduced [75]. Lower counts observed during the critical stage tend to increase in mild and severe conditions (characterized by a higher number of monocytes) (Figure 2), along with a reduced expression of inflammatory markers (CD38, sCD14, CRP, CD163, and soluble tissue factor) and an increased expression of HLA-DR$^+$ [42,75,76]. The convalescent stage, characterized by a stabilization of symptoms, has been described with a significantly increased level of monocytes, when compared to COVID-19-infected and healthy patients [21]. However, the participation of subtypes is still controversial [75,76,126]. Monocyte stabilization (and its subtypes) was demonstrated to occur only 151 days after infection [42,75].

On the other hand, macrophages are influenced by antibody production against SARS-CoV-2 epitopes. Polarization to M1 macrophages is also observed in convalescents, especially stimulated by IFN-$\gamma$, resulting in a further production of CXCL8 and CCL2, which are both important chemokines for monocyte migration to tissue [79,91,98,139]. IFN-$\gamma$ was described to play an important role in monocyte activity, with delayed IFN production (during acute disease) potentially driving a late inflammatory response against the virus [73]. Although inflammation is mainly driven by *M*1 macrophages, *M*2 macrophages

may also contribute with proinflammatory cytokines when antibodies are produced. IgG anti-SARS-CoV-2 is recognized by a complex of receptors, but inflammation is driven by FcγRIIa recognition. Typically, this receptor, together with other FcγRs, is largely expressed in macrophages; when stimulated, it contributes to a cytokine storm through Il-1β, IL-6, TNF-α, and CXCL8 production [140]. This highlights that, depending on the stimulus of adaptive immunity and further antibody titer in acute conditions, the immune dynamics may worsen disease progression (Table 1).

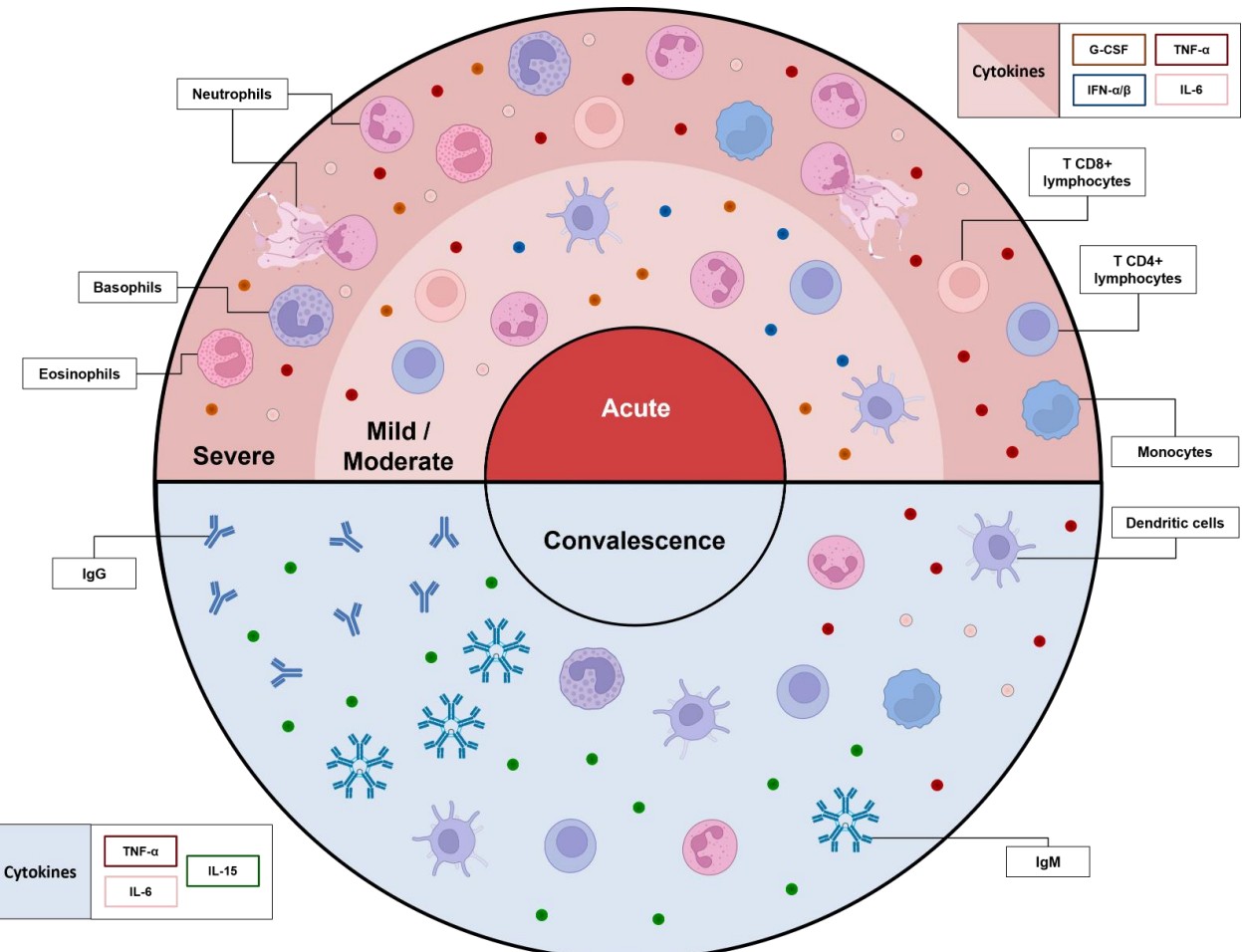

**Figure 2.** Transient immunological response in mild/moderate, severe, and convalescent patients with cellular and molecular profile. Severe patients experience intensive immune dynamics, driven by cytokine storm (TNF-α, IL-6, IFN-γ, and CCL2), which may worsen clinical condition, as well as compromise adaptive immunity. With clinical improvement, the Th2 profile becomes more prominent, with participation of IL-10, IL-2, IL-4, IL-13, IL-15, IL-18, CCL3, CCL4, and CCL2; however, if a Th1 profile persists, a poor outcome can be seen. Those with a mild/moderate profile have more antiviral properties (IFN-α and IL-15RA), with a higher proportion of T CD8+ lymphocytes, neutrophils (especially bands), and dendritic cells. Regulatory and anti-inflammatory mechanisms prevent severe conditions, but may also interfere during convalescence. This stage is marked by an improvement in monocyte count (for those severe) and a change in cytokines to a repairment profile. In convalescent patients, higher antibody IgM titers are seen during the first months after infection, while IgG may survive for years, potentiating the prevention of new infections. Adaptive cell surveillance prevails over innate immunity; however, local damage to tissues may cause further complications or even increase the risk of death.

**Table 1.** Major functions of innate immune cells and their subtypes during COVID-19 disease and convalescence. Phenotypic and functional dynamics are described, which may contribute to laboratorial and clinical aspects of SARS-CoV-2 infection, as well as participation during convalescence.

| Immune Cells | | COVID-19 | Performance in Convalescents | References |
|---|---|---|---|---|
| Neutrophils | | • Important marker of prognosis when calculated with ALC on neutrophil-to-lymphocyte ratio (NLR)<br>• Stimulus for NET release<br>• Increase in thrombosis risk and organ damage in severe cases<br>• Immature neutrophil (CD10$^+$) participation | • Reduction in NET production<br>• Senescent neutrophil appearance<br>• Low expression of activation markers on cell surface | [12,25,27,44,45,54,56] |
| Eosinophils | | • Protective against severe outcome in ICU patients<br>• Increased rate is directly proportional to patient's clinical improvement<br>• Lower in severe patients and with cytokine storm syndrome<br>• Activated neutrophils (CD69$^+$) related to poor outcome | • Contribution to higher titer of antibodies<br>• Low rate of infected eosinophils by SARS-CoV-2 | [9,28,36,68,69,138] |
| Basophils | | • High producer of inflammatory mediators (IL-3) | • Indirect contribution to antibody production through IL-4 and IL-6 production | [51] |
| Monocytes | Classical | • Lower | • Higher than in acute phase<br>• Higher expression of HLA-DR | [21,23,73,75,76,80] |
| | Inflammatory | • Higher level<br>• Associated with IL-6, TNF-α, and inflammatory markers in critical patients<br>• May be used to differentiate mild and severe patients | • Reduced levels<br>• Higher control of inflammatory status<br>• Stabilization of blood count 10 to 150 days after symptoms onset | |
| | Patrolling | | | |

**Table 1.** *Cont.*

| Immune Cells | | COVID-19 | Performance in Convalescents | References |
|---|---|---|---|---|
| Macrophages | M1 | • Inflammatory profile (CD80$^+$)<br>• Guided by IL-6 and TNF-$\alpha$<br>• Less interference on viral replication, when compared to *M*2 macrophages, due to fewer acid granules<br>• Higher expression of CD11b and CD11c, and lower expression of CD68 and CD169 | • Increased participation by IFN-$\gamma$<br>• Intense production of CXCL8 and CCL2 | [88,91,98,139] |
| | M2 | • Alveolar macrophages (CD206$^+$) show higher dimension, together with M1<br>• Interference with viral replication due to acid granules<br>• Higher expression of CD11b, CD11c, and CD169, and lower expression of CD68 | • Recognition of IgG anti-SARS-CoV-2 occurs by Fc$\gamma$RIIa and induces pro-inflammatory molecules (IL-6, IL-1b, TNF-a, and CXCL8). | [73,90,98,140] |
| Dendritic cells (DCs) | Plasmacytoids | • Lower in infected patients<br>• Proportional to a worse prognosis<br>• Reduced ability to produce IFN-$\alpha$ and TNF-$\alpha$<br>• Moderate expression of CD38 | • Increased, when compared to severe patients<br>• Suggested increase after diagnosis | [51,77,81,89,105] |
| | Monocyte-derived/myeloid DCs | • Lower in severe patients<br>• Low HLA-DR expression in severe patients<br>• Few functional alterations<br>• Increased expression of CD11b and CD11c, especially on CD1c$^+$ mDC | No material found | [12,86,88,105] |

The pDC and myeloid DC (mDC) counts during the acute phase are lowered in circulating blood, when compared to a convalescent and healthy status, related to the patient's age and potentially the migration of these cells from blood to lymph nodes [12,42,51,77,85,86,106,134]. Lower counts are seen in severe when compared to mild patients [12]. However, when compared to the convalescence stage, the total DC count remains lower when compared to normal individuals. Among subpopulations, only transitional DC had an increased rate [89], whereas CD1c[+] mDC was reduced, and CD141[+] and CD16[+] mDC exhibited no difference from healthy patients.

The participation of DCs is yet to be unraveled, but the damage occurring in severe patients prevails during convalescence. The cytokine storm experienced by some patients allows a greater effect of the negative IL-6/DC axis and further disease progression [106].

Convalescence is also marked by an improvement in ALC, which occurs rapidly in mild/moderate patients, compared to critical patients [82]. Furthermore, a Th1 profile is maintained, with higher prevalence of MAIT cells (CXCR3[+]), in those who had mild disease [136]. The reactive ability of MAIT cells (CD69[+]) remains for a few days after symptom onset, but reduce after 1 month [79]. Few studies evaluated T$\gamma\delta$ cells in convalescence, but a nonsignificant difference was demonstrated among healthy, infected, and convalescent patients [49,126]. Those with a severe form presented greater granzyme K production in convalescence, as well as a strong decrease in CD56[low]CD16[+] effector cells [116]. Those who presented a better outcome had normalization of helper, cytotoxic, and memory cells, but a perseverance of nonconventional Th1 cells, as well as an increase in IFN-$\gamma$, TNF-$\alpha$, and IL-10 cytokines. There is typically an improvement in parameters, but a long time is needed to return to normality [127,129].

Memory and follicular T cells and antibodies specific to spike proteins are low in the first 4 months after viral clearance in mild patients [132,141]; however, a T-cell response has been observed even after 6 months toward the spike, nucleocapsid, and M proteins [142]. Although B cells are stimulated following infection, few studies have explored the components underlying this activation. An increase in circulating B lymphocytes (CD19[+]CD10[+]) was shown to better drive adaptive immunity in symptomatic and RT-PCR-negative patients than in severe RT-PCR-positive patients [6,117,127,143] (Table 2).

**Table 2.** Participation of phenotypic and functional lymphocyte cells during COVID-19 disease and convalescence.

| Immune Cells | | COVID-19 | Performance in Convalescents | References |
|---|---|---|---|---|
| T lymphocytes | Helper (CD4+) | • Negative correlation with prognosis<br>• Contribution to cytotoxicity by T CD8+ cells<br>• Regulation of a Th1 polarized response and B-cell proliferation | • Specificity to spike, M, N, and RBD antigens for periods over 5 months | [6,128,132,144–147] |
| | Cytotoxic (CD8+) | • Increase in diagnosed patients (despite clinical presentation)<br>• Lower in severe patients<br>• Reduced frequency of memory and effector T CD8+ cells (CD45RO+ and CCR7−)<br>• Activated cytotoxic T cells (HLA-DR) do not show significant variation | • Severe patients show perseverance on production of granzymes, which might be related to long-COVID-19Specificity to viral antigens 4 weeks after clinical improvement | [5,20,105,116,132,147] |
| B lymphocytes | | • Increased circulation of B lymphocytes CD19+ in severe conditions | • Reactive memory phenotype against S, RBD, and N antigens<br>• Low seroreactivity rate to most antigens after periods over 6 months, but good (although reduced) seroreactivity rate to S antigen | [148,149] |

Atypical memory B cells were the first lineage of adaptive immunity described, and their perseverance might indicate a worse outcome. Instead, immature transitional cells are related to clinical improvement and a higher surveillance rate. In convalescence, a higher participation of classical memory cells has been found, with a more prominent involvement of immature transitional B cells [150]. Immune reactivity to SARS-CoV-2 was observed even 5 months after infection, especially to spike, RBD, and N proteins [132,148].

Nonconventional T cells demonstrated reactivity and expansion even 3 months after the first SARS-CoV (in 2003) infection, which was further correlated to IgG anti-SARS-CoV [151]. Analyses have demonstrated that antibodies participate in immune defense, even though an increase in effector cells suggests a better clinical outcome in those patients diagnosed with COVID-19 [135]; nonregulated production of antibodies may cause a hyperactive immune response and lead to further immune cell infiltration into the lungs [140].

The majority of B lymphocytes are characterized as memory cells [116] and $CD27^-$ $IgD^-$ [127], which mainly produce IgM a few weeks after viral clearance, as well as IL-10. Plasmablast participation is transient during the convalescence of patients who experience mild disease, but sustained in severe cases [39,48,89,116]. Transitional cells ($CD24^{high}CD38^{high}$) markedly increased in the transition from severe disease to convalescence, whereas mild and moderate patients showed no significant difference. Ki-67, an important protein for cell proliferation, is increased in acute patients, when compared to convalescence, showing a positive correlation with T helper cells [85,127], which might be due to the intense stimulus during the acute phase, with a further regulation process in convalescence [143]. In contrast, a discussion of viral infectivity must be established, as some studies described the participation of higher viral titers in severe patients, as well as the ability to effectively induce both innate and adaptive immunity [132].

Few data are available on adaptive immunity and the factors associated with the convalescent stage; furthermore, controversial data have been presented, hindering comprehension of the main component contributing to higher or lower production of antibodies. It is clear, however, that anti-spike lymphocytes have a low survival rate when compared to other SARS-CoV-2 antigens, whereas more studies must be conducted to determine the influence of dendritic cell activation and cytokine involvement to unravel this issue. These factors may contribute to immune efficacy during the acute stage and impair adaptive immunity activation. We recognize that this process may contribute to not only treatment protocols, but also the availability of neutralizing antibodies in convalescence.

Specific antibodies against COVID-19 proteins are key factors influencing immune protection. Many studies have found that those convalescent patients who experienced a symptomatic stage of COVID-19 have a higher rate of antibody production [152]. Those with fever, cough dyspnea, and pneumonia are 50 times more likely to produce higher antibody titers [138,153]. The presentation of higher titers of antibodies in severe patients has been discussed; although some studies have proposed that the severity of symptoms is related to a higher viral load and the availability of viral antigens to induce the immune system, only the anti-SARS-CoV-2 spike/RBD region antibody is increased in severe patients [85,132].

The antibody response is an important step in immunity that actively participates in cellular response, complement activation, and immunity protection [82,153]. One of the functionalities of antibodies includes the activation of the classical pathway of the complement system, thus mediating the damage to infected tissue and release of anaphylatoxins. Although aimed at viral clearance, it was identified that the immune complex from the membrane attack complex is fixed on lung vessels, potentially causing irreversible pulmonary damage in fatal COVID-19 [82].

A negative correlation has been found between lymphocyte count and antibody titers, suggesting that an intense acute inflammatory response may compromise antibody production during the convalescent stage [152]. Patients in the acute phase that produced higher IFN-γ levels were demonstrated to have higher antibody titers and greater lymphocyte activation, suggesting that adaptive immunity is IFN-γ-dependent [151]. However, viral

load during the acute phase was also highlighted as a possible interfering with antibody production [124].

Detectable IgM antibodies are found in less than 60% of convalescent patients 4 weeks after COVID-19, while IgG is detectable in more than 75% [148]. Moreover, no seroreactivity has been reported in convalescents who experienced reinfection [10]. Anti-nucleocapside IgG1 and IgG3 were found to increase along recovery, whether in critical or noncritical cases, reaching their peak 10 to 20 days post symptoms. In contrast, IgG2 and IgG4 were poorly detected [26,82].

The specificity of antibodies against different epitopes has been described. The main antibodies studied are against nucleocapsid (N), spike (S), and RBD proteins, in which a rapid decrease in IgG anti-N and anti-RBD has been observed, whereas anti-S is seemingly more stable in serum, which is further correlated to memory B cells and circulating Tfh cell survival up to 10 months [124,153–155]. All antibodies, despite reducing over time, exhibit an increase in their neutralization ability and their ability to bind to B lymphocytes [153,156]. Somatic mutations and defects in the germinal center at the beginning of convalescence were demonstrated to be related to antibody titers [156].

IgA has a serum conversion around 70% in noncritical patients and 100% in critical patients [82]. Some studies have suggested that IgA levels (specifically anti-RBD) tend to remain detectable in serum due to antigen presentation by follicular DCs and long-lived antigens [153,156]. In mild cases, IgA anti-N demonstrates a transient increase, with a peak 15 to 20 days post symptoms, whereas critical patients, despite a reduction, exhibited more stability until around 40 days [26,82].

Different strategies have been proposed to induce adaptive immunity. By 2022, many vaccines were developed; despite convalescence or vaccination, it has been observed that previously exposed patients still have a low risk of contracting COVID-19 [157,158]. These concerns, together with vaccine efficacy, remain to be addressed due to several issues: (1) individual immunization rate; (2) seroconversion; and (3) completeness of vaccination strategy. We do recognize that the immune response during the acute and convalescent stages of disease has an established profile, while the analysis of vaccinated and reinfected patients must also be fully reviewed to determine the factors predicting severity of new cases and to improve quality of life.

## 4. Conclusions

SARS-CoV-2 has infected millions of people around the world, being responsible for many deaths, as well as sequelae in those who survived. Many issues of immunity have been raised, mainly the challenge in responding to a new virus, the disbalance in immune response, the cytokine storm, and the hyperinflammation dynamics in severe patients, which can compromise further activation of the adaptive system. The pathway taken by the immune system in the acute phase is not only directly related to the outcome, but its effects can also be seen during the convalescence phase.

Although the participation of cells acts as a hallmark of disease progression and severity, neutrophils, lymphocytes, and eosinophils have been revealed as good predictors of clinical outcome in reaching convalescence. A remarkable 'shift to the left' is clear, which contributes to chemotaxis and impaired functionality in convalescence. Type I IFN and IL-15 are related to guide immunity, as well as thrombotic events, orchestrated by NETs and basophils.

The dynamics in the acute phase are coordinated by granulocytes and enzymes, alleviated by increased monocytes in convalescence. However, this functionality is situation-specific, and the participation of macrophages in tissues after viral clearance remains unclear.

We recognize that this review focused only on people infected with SARS-CoV-2 and not other diseases, such as HIV, lupus, diabetes, and immune disorders, which could change the immune profile and, thus, the patient's severity score. The dynamics of immunity interfere with the convalescent stage, and several questions have been raised since the emergence of SARS-CoV, related to viral biology, human immune response, and treatment

strategies; the appearance of SARS-CoV-2 has resulted in further questions. Several studies have proposed that convalescence, with no distinction of time, is a checkpoint for many cells and molecules.

Comprehending whether immune factors induce a potent response to mild and severe COVID-19, especially in convalescents, is essential for proposing novel treatments, increasing quality of life, and improving disease prognosis. Long immunization is now a priority to prevent the severity of new cases of COVID-19, as well as reduce the mortality rate of other wild viruses. Public policies and new strategies must be addressed under this framework, due to the chaotic situation experienced by several countries.

**Author Contributions:** Conceptualization, A.L.S.-J. and A.M.; investigation, A.L.S.-J., L.d.S.O. and N.C.T.B.; writing—original draft preparation, A.L.S.-J., L.d.S.O. and N.C.T.B.; writing—review and editing, A.L.S.-J., A.M.T., A.G.d.C. and A.M.; supervision, A.G.d.C. and A.M.; project administration, A.L.S.-J.; funding acquisition, A.M. All authors have read and agreed to the published version of the manuscript.

**Funding:** This work was funded by the Fundação de Amparo à Pesquisa do Estado do Amazonas (FAPEAM) (Pró-Estado Program (#002/2008, #007/2018, and #005/2019), the POSGRAD Program (#005/2022), the PCTI-EMERGESAÚDE/AM—Chamada II Program (#006/2020)), the Conselho Nacional de Desenvolvimento Científico e Tecnológico (CNPq), and the Coordenação de Aperfeiçoamento de Pessoal de Nível Superior (CAPES)/Consolidação doutorado. A.L.S.-J., L.d.S.O. and N.C.T.B. have fellowships from CAPES and FAPEAM (Ph.D. and SI students). A.G.d.C. and A.M. are level 2 research fellows from CNPq. A.G.d.C. is a research fellow from FAPEAM (PRODOC Program #003/2022). The funders had no role in study design, decision to publish, or preparation of the manuscript.

**Institutional Review Board Statement:** Not applicable.

**Informed Consent Statement:** Not applicable.

**Data Availability Statement:** The data used to support the findings of this study are included within the article.

**Acknowledgments:** The authors would like to extend their gratitude to all laboratory staff for their support of this research.

**Conflicts of Interest:** The authors declare no conflict of interest.

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
