# Peer review of "Immune Dynamics Involved in Acute and Convalescent COVID-19 Patients"

_2673-5601, doi:10.3390/immuno3010007_

Round 1
Reviewer 1 Report
The authors have submitted the review article entitled ‘COVID-19: An Immune Review of Acute to Convalescent Patients’. This article is well written and have described in detailed the changes in immune cell functions of Covid-19 patients. The authors have discussed sequentially the impact of acute to post-covid on immune cell functions. There are various typos, grammatical and structural mistakes that needs revision.
What is the difference between ‘post-acute COVID-19, and post-COVID-19 syndrome. Please state in the briefly in the article. Is there a difference between post-covid syndrome and Long-covid. Please clarify
There are several grammatical and spelling errors in the manuscript. Please take a look carefully as some sentences does not make any sense. For example…
Line 22., abstract section. The sentence ‘comprehension of convalescent patients is still a lack of discussion is not clear. The meaning is not clear.
Line 474-475, ‘Usually are 474 those patients who experienced COVID-19 (symptomatic or not) but had not involved to 475 death’. The meaning is not clear.
Line 694-695., ‘Even though the participation of cells acts as hallmarks of disease progression and severity, the interplay of neutrophil, lymphocytes and eosinophils have shown a good predictor on clinical outcome in patients with the severe’. Revise the sentence
Reviewer 2 Report
The review article by Silva-Junior et al presents a discussion on the immune response to COVID-19 in the acute phase and in convalescent patients. It covers information from many different studies across the globe and puts focus on discussing how the immune response can be utilized as markers for disease severity and prognosis. The topic is very relevant at the present time and the idea and effort of connecting immune response from the acute phase up to convalescence under one discussion are interesting. However, there are some concerns about the review as listed in the comments below.
Major comments:
1. The entire review requires thorough proof reading as well as editing. There are many grammatical errors throughout the article, especially in using prepositions and conjunctions. Many of the sentences are too long and convoluted (some specifically marked in the minor comments) which reduces the comprehensibility of the message.
2. While the review covers many literature reports from 2020 and 2021, I found coverage of the more recent reports from 2022 lacking in the manuscript.
3. The ‘Overview of immunology’ section is very long. It would benefit from division into subsections focusing on individual cell types. Also, though the section is titled overview of immunology, the focus is primarily on immunological markers. The authors might consider a more specific title for this section.
4. A tabular representation of markers with COVID-19 severity and convalescence would benefit the review.
5. The word ‘seems/seemed to’ is often used in the manuscript. It suggests a lack of confidence in the statement made by the author or the report cited. The authors should consider changing the tone of these sentences.
6. Long COVID-19 has emerged as a big threat in convalescent patients. It is also mentioned among the keywords in the manuscript. But there is very little discussion on this condition. There needs to be a larger section focused on the known immune hallmarks and markers in long COVID-19 cases.
7. Despite a significant focus on antibody response to COVID-19 and convalescent patients, the effects of COVID-19 vaccination and reinfection are absent from the manuscript.
8. A discussion on the knowledge gaps in immunological markers in different aspects of COVID-19 prognosis would be valuable.
9. It would be interesting if the authors put forward any original suggestions or ideas on immunological markers of COVID-19 to help clinical decision-making as a result of their effort in consolidating a large number of studies.
Minor Comments:
1. Abstract (line 15): “Participation of immunity showed as a major factor…”
2. Abstract (line 21): ”COVID-19 was very explored along time, however, look for new perspectives under immunology must be highlighted, and comprehension of convalescent patients is still a lack of discussion.” – Unclear, needs rephrasing.
3. Line 29-31: The causal pathogen for COVID-19 is SARS-CoV-2 (full form severe acute respiratory syndrome coronavirus 2). Why is it written as ‘related to coronavirus type 2’?
4. Line 32-34: Although there are reports suggesting presence of SARS-CoV-2 virus and viral RNA in in clinical samples and fomites, the extent of transmission through those routes have been under question (Ref: Goldman, E. Lancet Infect. Dis. 20, 892–893 (2020)., Ben-Shmuel, A. et al. Clin. Microbiol. Infect. 26, 1658–1662 (2020), Lednicky, J. A. et al. Int. J. Infect. Dis. 100, 476–482 (2020)). Therefore it needs to be more explicitly mentioned that these different modes are possible routes of transmission that still need conclusive evidence and the primary mode of transmission is via aerosol.
5. Line 46: “…severe patients, led by a cytokine storm…”
6. Line 47-49: There are now multiple studies on the immune response of convalescent patients, eg: Zhao et al, 2022, Int Immunopharmacol, Pan et al, 2021, Signal Transduction and Targeted Therapy, Garanina et al, 2022, Frontiers in Microbiology etc. These should be mentioned and cited here.
7. Line 81-85: Sentence is too long and convoluted.
8. Line 95-97: Meaning not coming out clearly. Needs rephrasing.
9. Line 100-104: Sentence is too long and convoluted.
10. Line 155: “…are a way out of infection..”, seems misplaced in the sentence.
11. The general discussion on NETs is too long. It could be shortened to directly focus of NETs in COVID-19.
12. Line 175 and 180: ‘under’ should be replaced with ‘in’.
13. Line 179: Full forms of MPO and NE-DNA need to be provided.
14. Line 189-191: Requires reference for the statement.
15. Line 232-239: Are the three hypotheses mentioned derived from the same paper cited in number 63, or are these hypotheses put forward by the authors? What is signified by ‘hypothesis not fully understood yet’?
16. Line 274: What is meant by “A concise discussion must be made here, once basophils….”? I did not see any further discussion on this topic following this statement.
17. Line 278: The term from monocytes (and lymphocytes) is ‘agranulocyte’, not ‘no-granulocyte’.
18. Line 289: ACE2 and its function as the primary receptor for SARS-CoV-2 is not mentioned anywhere before this sentence. Its relevance will not be clear to a reader unaccustomed to COVID-19 biology.
19. Line 293: Why write ‘however’ in this sentence? These two parts do not seem to counter each other.
20. Line 300-303: Meaning of the sentence is unclear.
21. Line 325: typological error in the word ‘established’.
22. Line 370: DC3 cells need to be explained.
23. Line 387-389: Sentence is convoluted, needs rephrasing.
24. Line 393: What is meant by ‘normalization level’? Did the authors mean normal levels?
25. Line 396: What B7-1 and B7-2 are needs explanation. Why their role should be explored also need an explanation.
26. Line 406: “…who involved to death” needs rephrasing.
27. Line 421-423: Sentence requires rephrasing to correct grammar.
28. Line 425: Why use ‘instead’? Few cytokines are related to what outcome?
29. Line 430: ‘..but also...’ should be replaced by ‘and also’.
30. Line 435-437: Meaning of the sentence is unclear.
31. Line 449: ‘unbalance’ should be replaced by ‘imbalance’.
32. Line 474: The sentence “Usually those …” is grammatically wrong and incomplete.
33. Line 547-549: This sentence does not follow through from the rest of the paragraph. How immunostimulatory or immunosuppressive effect is regulated by stimulus amount is not explained or referred to in other literature.
34. Line 601-606: Sentence is convoluted, meaning unclear.
35. Line 632: “…but we believe..” What is the basis of this declaration?
36. Line 689-691: Why is immune underpreparedness mentioned in contrast to cytokine storm and immunological dynamics? The immune system being under-prepared to deal with a new virus often leads to a disbalance in immune response and hyperinflammation. This holds validity in Covid-19 as we saw a marked decrease in disease severity in vaccinated individuals or upon reinfections, which prepared the immune system to recognize the virus better.
37. Line 692-693: Meaning unclear.
38. Line 704-706: How do the authors expect other diseases to affect the immune profile in COVID-19 patients and in convalescence? The authors mentioned some immune disorders here. How will other coinfections and comorbidities affect similar parameters?
39. Line 706-708: Which questions and points are the authors referring to? It must be better explained.
40. Line 713: ‘Public politics’ or ‘Public policies’?
Round 2
Reviewer 2 Report
I thank the authors for incorporating the suggested modifications and I believe the manuscript is vastly improved now. I have two minor comments on the present manuscripts as listed below.
1. Line 39-41: The expression levels of ACE2 are known to be only moderate in the respiratory tract, not high (as observed in the intestine and kidney). ACE2 expression gets significantly higher as a response to SARS-CoV-2 infection. It would be best not to phrase the sentence as ‘vulnerable due to the higher expression…’
2. The section title of 2.3 should be rephrased into a non-interrogative sentence.
Author Response
Dear revisor,
We thank for the comments. We have made the proper changes to improve comprehension, and the new document is attached, with changes highlighted.
Topic 2.3 was restructured, and also topic 3, which was in an interrogative format.